# Local Geometry Constraints in V1 with Deep Recurrent Autoencoders

**Jonathan Huml**
School of Engineering and Applied Sciences
Harvard University
Cambridge, MA
jhuml@g.harvard.edu

**Abiy Tasissa**
Department of Mathematics
Tufts University
Somerville, MA

**Demba Ba**
School of Engineering and Applied Sciences
Harvard University
Cambridge, MA

## Abstract

Sparse coding is a pillar of computational neuroscience, learning filters that well-describe the sensitivities of mammalian simple cell receptive fields (SCRFs) in a least-squares sense. The overall distribution of SCRFs of purely sparse models, however, fail to match those found experimentally. A number of subsequent updates to overcome this problem limit the types of sparsity or else disregard the dictionary learning framework entirely. We propose a weighted $\ell_1$ penalty (WL) that maintains a qualitatively new form of sparsity, one that produces receptive field profiles that match those found in primate data by more explicitly encouraging artificial neurons to use a similar subset of dictionary basis functions. The mathematical interpretation of the penalty as a Laplacian smoothness constraint implies an early-stage form of clustering in primary cortex, suggesting how the brain may exploit manifold geometry while balancing sparse and efficient representations.

## 1 Introduction

Overcomplete sparse coding as a model of the primary visual cortex (V1) is a pillar of computational neuroscience (Olshausen & Field, 1997). Training on natural image patches[1] via a Hebbian learning rule produces filters that are spatially localized, bandpass, and oriented to a select range of rotation angles. These filters are similar to those observed in the mammalian cortex (Jones & Palmer, 1987), which are well-described by two-dimensional Gabor functions. However, the properties of sets of Gabor filters fitted to the simple cell receptive field (SCRF) estimates produced by sparse coding have been shown to misalign with filters fitted to rhesus macaque responses to drifting sinusoidal gratings (Ringach, 2002) as a distribution (rather than an individual, least-squares sense). In particular, the original sparse coding (SC) model overpredicts and underpredicts the frequency of well-tuned and broadly-tuned cells, respectively. Well-tuned cells maintain several (more elongated) subfields than the "blob-like" broadly tuned cells, as shown in Figure 1. An animal with such a set of computationally-learned filters might not be able to detect certain edges or shapes, and might use more filters to represent what could otherwise be represented with fewer given a larger diversity of SCRF shapes.

A number of models have been subsequently proposed as a result. Of particular interest, (Rehn

---

[1]http://www.rctn.org/bruno/sparsenet/IMAGES.mat

4th Workshop on Shared Visual Representations in Human and Machine Visual Intelligence (SVRHM) at the Neural Information Processing Systems (NeurIPS) conference 2022. New Orleans.

& Sommer, 2007) limits the number of active neurons rather than the average neural activity, which significantly improves diversity of shapes. (Zylberberg et al., 2011) develops a spiking network based on synaptically local information to overcome this discrepancy.

Hypothesizing that explicit image reconstruction is not a biologically relevant task, (Yerxa & Simoncelli, 2022) proposes a novel contrastive objective, Local Low Dimensionality (LLD), that minimizes the dimensionality of encodings of spatially local image patches relative to their global dimensionality.

While LLD diversifies SCRF shapes compared to sparse coding, (Shen et al., 2019) uses deep methods to reconstruct the brain's perceptions of images from functional magnetic resonance imaging data, showing the relevance of reconstructive models of the visual cortex. We also maintain that while reconstructive frameworks may not be exactly the objective function of the brain, they may be computational stand-ins for artificial systems rather than one-to-one recreations (e.g. back-propagation). In addition to past success of the reconstructive

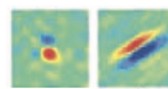

Figure 1: Broadly-tuned (left), well-tuned (right) macaque SCRFs Ringach (2002)

framework updates, we therefore investigate a deep recurrent autoencoder architecture with additional regularization constraint to enforce a similar flavor of locality: namely, a weighted-$\ell_1$ penalty (WL) that encourages latent representations to use a specialized set of local neurons. While the hierarchical setting is left to future work by LLD, this architecture naturally learns a hierarchy of representational units when trained with an additional discriminative loss term (Rolfe & LeCunn, 2013). The reported findings motivate the need for *spatial regularization* of neurons and necessitate more precise arguments against reconstructive frameworks like sparse coding.

## 2 Previous Work

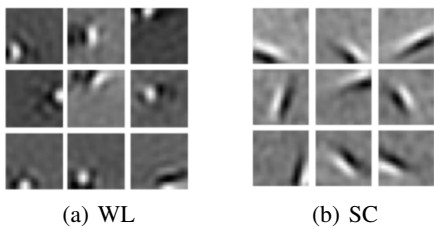

(a) WL          (b) SC

Figure 2: The weighted-$\ell_1$ penalty begets more of the "blob-like" SCRFs that are missing from the original model.

The general neural coding framework can be formulated as:

$$\mathcal{L}(\mathbf{A}, \mathbf{X}) = \frac{1}{2}||\mathbf{Y} - \mathbf{A}\mathbf{X}||_F^2 + S_\lambda(\mathbf{X}) \tag{1}$$

where $\mathbf{Y} \in \mathbb{R}^{d \times n}$ is a set of $n$ stimuli of dimension $d$, $\mathbf{A} \in \mathbb{R}^{d \times m}$ is a learned set of $m$ basis functions, $\mathbf{X} \in \mathbb{R}^{m \times n}$ is a set of $n$ latent representations of inputs, and $S_\lambda(\mathbf{X})$ is a regularization penalty. (Rozell et al., 2007), among other advances, associates sparse coding with $S_\lambda(\mathbf{X}) = ||\mathbf{X}||_1$ (columnwise). However, while neurons fire sparsely, they are also specialized to certain types of visual stimuli in the input space. As formulated, (1) makes no explicit assumptions about the structure of this sparsity in neural latent space.

Nonetheless, the filters learned on natural image patches are quantitatively well-described (in a least-squares sense) by two-dimensional Gabor functions for $(x, y) \in \mathbb{R}^2$, where:

$$G(x', y') = K \exp\left(-\left(\frac{x'}{\sqrt{2}\sigma_{x'}^2}\right)^2 - \left(\frac{y'}{\sqrt{2}\sigma_{y'}^2}\right)^2\right) \cos\left(2\pi f x' + \phi\right) \tag{2}$$

and $(x', y')$ are obtained from a rotation of angle $\theta$ and translation $(x_0, y_0)$.
In our experiments, these eight parameters are fit through a gradient descent scheme that alternates over the parameters while holding the others fixed. The filters are fit on $N = 100,000$ randomly extracted patches (of varying sizes, as explored in the Appendix) on the original Sparsenet data (Olshausen & Field, 1997). We also began to fit on the CIFAR10 dataset (Krizhevsky et al., 2010) in anticipation of future discriminative tasks. While the sparse coding filters are well-described by Gabor filters in a least-square error sense, the learned Gabor parameters misalign with primate data as shown in Figure (3).

As discussed in the introduction, many different approaches can be taken to address this problem.

Specifically, we focus on local computation enforced via the objective function. LLD discards the reconstruction loss and encodes natural image stimuli to local ensembles of image patches $\{(\mathbf{x}_1^{(1)}, \ldots, \mathbf{x}_n^{(1)}), \ldots, (\mathbf{x}_1^{(B)}, \ldots, \mathbf{x}_n^{(B)})\}$, with superscripts denoting local ensembles and subscripts denoting ensemble members. LLD is a shallow network where $\mathbf{s}_i^{(j)} = \text{ReLU}(\mathbf{W}\mathbf{x}_i^{(j)} + \mathbf{b})$. A covariance matrix $\mathbf{\Sigma}_l^{(j)} = \text{Cov}\left([\mathbf{s}_1^{(j)}, \ldots, \mathbf{s}_n^{(j)}]\right)$ is formed on the $j^{th}$ ensemble. The LLD loss is formulated as:

$$\mathcal{L}(\mathbf{W}, \mathbf{b}) = \frac{\mathbb{E}_j[\text{tr}(\mathbf{\Sigma}_l^{(j)})]}{\text{tr}(\mathbf{\Sigma}_g)} \tag{3}$$

where $\mathbf{\Sigma}_g$ is the response covariance to all patches in the batch. The model is able to better replicate the diversity of SCRF shapes, which exhibit a phase symmetry in rhesus macaque data.

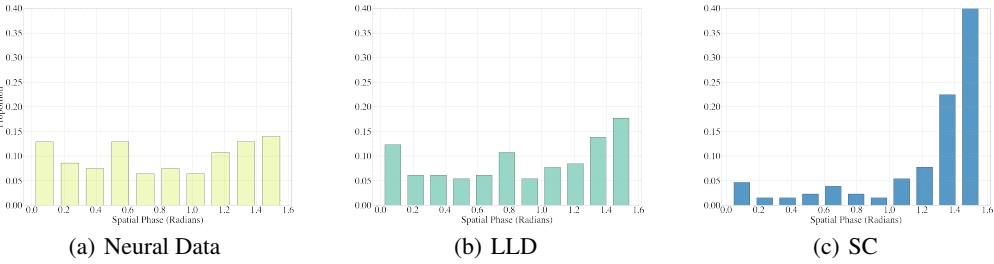

(a) Neural Data      (b) LLD      (c) SC

Figure 3: Gabor spatial phases of rhesus macaques are largely bimodal, but SC phases are highly skewed due to the absence of "blob-like" fields in Figures 1 and 2.

## 3 Locality-Constrained Reconstructive Frameworks

Can additional structure be incorporated into the reconstructive framework to better match the experimental data? This question may be addressed through additional regularization to encourage local representation, taking inspiration from concepts of locality used in manifold learning, especially spectral methods and locally linear embedding (Roweis & Saul, 2000). The weighted-$\ell_1$ constraint penalizes neural encoding activity based on the distance between the natural image stimuli and the basis functions $\mathbf{a}_j$, where:

$$S_\lambda^{WL}(\mathbf{X}) = \mathbb{E}_{i \in [n]} \left[ \sum_{j=1}^m x_j ||\mathbf{y}_i - \mathbf{a}_j||_2^2 \right] \tag{4}$$

Here, neurons are specialized to certain types of input as large neural energy requirements will limit the strength of the firing rate $x_j$. In contrast to the original sparse coding alternating minimization scheme, this objective is solved through algorithm unrolling (Monga et al., 2020) into a deep recurrent autoencoder, which projects the encodings onto the probability simplex through a nonlinearity $\mathcal{P}_S$ described below.

Mathematically, one might recognize the weighted-$\ell_1$ penalty as a Laplacian quadratic form of a graph. Which Laplacian and graph, specifically? This can be formulated as a bipartite graph Laplacian on the $m + n$ basis functions and inputs (vertices), whose edge weights between the $y_i^{th}$ and $a_j^{th}$ vertices are $x_{ij}$, and 0 otherwise (i.e. no self-loops). Thus, the stimuli can be easily clustered by performing an eigendecomposition on this constructed Laplacian. In a discriminative classification task, this will allow for a more rigorous analysis of a given neuron's sensitivity to various class types.

One potential criticism of the autoencoder architecture used here (and deep learning in general) is the lack of plausibility for the back-propagation algorithm being implemented by the brain (Bengio et al., 2016). While this criticism is fair, the presented results do not attempt to form a one-to-one map with computation in V1, but rather attempt to rectify a non-plausible learning mechanism with geometric regularization and a procedure that learns hierarchical representations. Shallow networks have long been commensurate with local, Hebbian update rules, while the expressivity associated with depth is indeed one of the more enigmatic features of modern neural networks and their high performance on many different tasks. However, while most deep architectures are "information soups"

where the parameters do not necessarily maintain any particular meaning, the weights of the unrolled network correspond exactly with $\mathbf{A}$ and $\mathbf{X}$, lending an interpretability aspect to the architecture not present for most of the top-performing networks today.

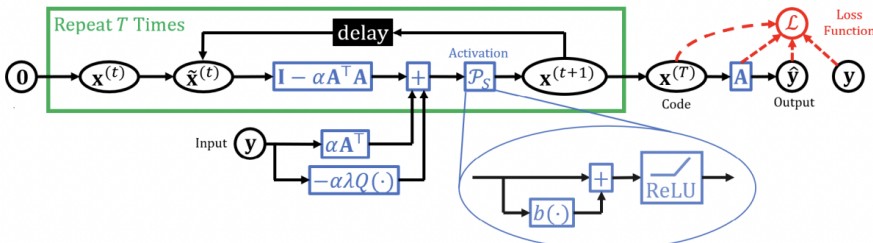

Figure 4: The unrolled architecture that learns $\mathbf{A}$ and $\mathbf{X}$, where $\mathcal{P}_S = \text{ReLU}\left(\mathbf{x} + b\left(\mathbf{x}\right) \cdot \mathbf{1}\right)$ and $Q\left(\mathbf{y}\right) = \sum_{j \in [m]} ||\mathbf{y} - \mathbf{a}_j||^2$ is a quadratic neuron. See appendix for details (Tasissa et al., 2021).

We have also explored an iterative Laplacian scheme (Kodirov et al., 2015) where:

$$S_\lambda^{LAP}(\mathbf{X}) = \text{tr}(\mathbf{X}\mathcal{G}\mathbf{X^T}) \tag{5}$$

for a pre-constructed (or iteratively updated) graph Laplacian $\mathcal{G}$. The penalty (5) is typically used in addition to an $\ell_1$ penalty. Here, however, $\mathcal{G}$ is built on the stimuli space to preserve local pairwise distances in latent space, whereas the weighted-$\ell_1$ penalty essentially interpolates the manifold in $\mathbb{R}^d$ with the set $\mathbf{a}_{j \in [m]}$ and then uses as few basis functions as possible. The Laplacians are fundamentally different. Thus $S_\lambda^{LAP}$ only constrains firing rates, while $S_\lambda^{WL}$ constrains both the firing rates *and* the learned basis functions, which we refer to as "spatial regularization."

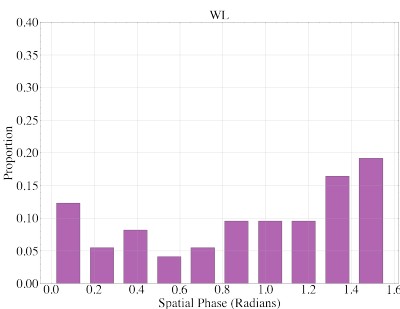

Figure 5: Spatial phases of the weighted-$\ell_1$ (WL) autoencoder. Locality-regularization is able to shift the original sparse code distribution of spatial phases

In our experiments, we find that the additional weighted-$\ell_1$ regularization technique shifts the spatial phase distribution to a more diverse range and vastly improves symmetry. The weighted-$\ell_1$ penalty makes the general sparse coding framework competitive with other local frameworks like LLD in terms of symmetry while maintaining the core ideas of the original model.

## 4   Conclusion

The improved spatial symmetry warrants further exploration into deep recurrent autoencoders (with varying flavors of locality constraints) as a model of the primary visual cortex. Is *explicit* image reconstruction biologically plausible? This assumption may be loosened in future work by considering a distribution of codes instead of a point estimate (Park & Pillow, 2020). However, given previous work showing the intrinsic hierarchical structure of discriminative recurrent sparse autoencoders (Rolfe & LeCunn, 2013), the findings presented here offer a potential path towards rigorously describing the differentiation of receptive fields that match experimental data.

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

# A  Appendix: Unrolled Network and Training

The spatial phase plots are obtained through a three-step process:

1. Train the (unrolled) network on the original Sparsenet data (Olshausen & Field, 1997) image patches
2. Using the learned basis functions, obtain the simple cell receptive field estimates through spike-triggered averaging
3. Fit the 2D-Gabors to the the receptive field estimates

For step (1):

**Encoder:**

Let $\mathbf{A}_{ij}^{(0)} \sim \mathcal{N}(0,1)$ and $\mathbf{x}^{(0)} = \tilde{\mathbf{x}}^{(0)} = \mathbf{0}$

Then, given $\mathbf{A}$ and $\mathbf{y}$, we solve for $\mathbf{x}^* \in \mathrm{argmin}_{\mathbf{x}} \mathcal{L}(\mathbf{A}, \mathbf{y}, \mathbf{x})$ by projected gradient descent:

$$\mathbf{x}^{(t+1)} = \mathcal{P}_S \left( \tilde{\mathbf{x}}^{(t)} - \alpha \nabla_{\mathbf{x}} \mathcal{L}(\mathbf{A}, \mathbf{y}, \tilde{\mathbf{x}}^{(t)}) \right) \tag{6}$$

$$\tilde{\mathbf{x}}^{(t+1)} = \mathbf{x}^{(t+1)} + \gamma^{(t)}(\mathbf{x}^{(t+1)} - \mathbf{x}^{(t)}) \tag{7}$$

for $t \in [T]$. In the code, we run $T = 15$ iterations of projected gradient descent (similar to FISTA). We have $\alpha = \sigma_{\max}(\mathbf{A})^{-2}$ and $\gamma^{(t)}$ is given by:

$$\gamma^{(t)} = \frac{\eta^{(t)} - 1}{\eta^{(t+1)}}, \qquad \eta^{(t+1)} = \frac{1 + \sqrt{1 + 4\eta^{(t)}}}{2}, \qquad \eta^{(0)} = 0 \tag{8}$$

The gradient of the weighted-$\ell_1$ penalty is given by:

$$\nabla_{\mathbf{x}} \mathcal{L}^{WL}(\mathbf{A}, \mathbf{y}, \mathbf{x}) = \mathbf{A}^T(\mathbf{A}\mathbf{x} - \mathbf{y}) + \lambda \sum_{i=1}^{m} ||\mathbf{y} - \mathbf{a}_j||^2 \mathbf{e}_j \tag{9}$$

We also explored a Laplacian penalty $S_\lambda^{LAP}(\mathbf{X}) = \mathrm{tr}(\mathbf{X}\mathcal{G}\mathbf{X}^{\mathbf{T}})$ to promote locality. The gradient of this penalty is most clean when written in a batch setting:

$$\nabla_{\mathbf{X} \in \mathbb{R}^{m \times b}} \mathcal{L}^{LAP}(\mathbf{A}, \mathbf{Y}, \mathbf{X}) = \mathbf{A}^T(\mathbf{A}\mathbf{X} - \mathbf{Y}) + \lambda \left( \mathbf{I}_{b \times b} + \mathbf{X}(\mathcal{G}^T + \mathcal{G}) \right) \tag{10}$$

where $D - A = \mathcal{G} \in \mathbb{R}^{b \times b}$ is a graph Laplacian built from a binary $k$NN graph on the inputs $\mathbf{Y}_{n \times b}$; that is, the edge weight between $\mathbf{y}_i$ and $\mathbf{y}_j$ is 1 if $i, j$ are $k$-nearest neighbors and 0 otherwise. We choose $k = 4$ in our experiments, though more rigorous analysis is required to determine the effect of this hyperparameter.

**Decoder:**

The decoder is a simple linear readout, where given $\mathbf{A}$ and $\mathbf{x}$, $\hat{\mathbf{y}} = \mathbf{A}\mathbf{x}$

**Training details:**

We used one GeForce GTX 1080 Ti GPU for training. However, the computational requirements are fairly modest. All of the models can be ran without a GPU, even with a batch size of $16 - 32$ with several hundred filters in at most in $1 - 2$ hours on a 2.6 GHz 6-Core Intel Core i7 PC with 32GB of RAM.

Because we use the same dataset, we also used the same variance normalization procedure described in the original paper (Olshausen & Field, 1997) to tune $\lambda$, which is easily the most important hyperparameter. In our experiments, we found that the weighted-$\ell_1$ network was fairly robust to changes in $\lambda$ even over several orders of magnitude, while the original Sparsenet algorithm only produces Gabor-like filters over a small neighborhood around $\lambda \approx 0.01$. Below, for example, we show filters for $\lambda = 0.001$ where each model is ran until convergence. The unrolled network looks

far more like the results in Figure 2 (where $\lambda = 0.01$) while Sparsenet is not able to learn its classic filters (holding all else constant).

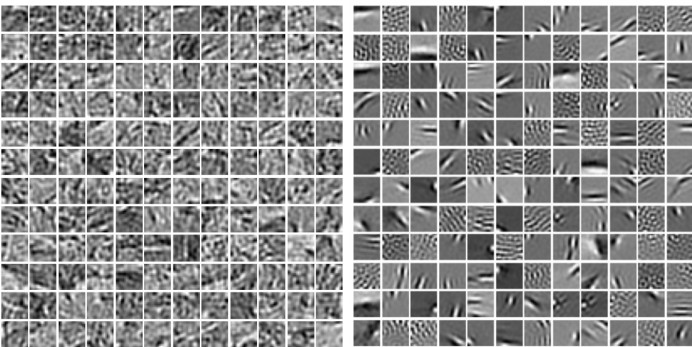

Figure 6: The Sparsenet algorithm (left) seems less robust to changes in $\lambda$ compared to the unrolled network (right).

# B    Appendix: Discriminative Task on Whole Images

Much to our surprise, the weighted-$\ell_1$ loss and unrolled architecture also seems to learn Gabor-like filters even on whole (albeit small) images in addition to random image patches. Below, we include a set of filters learned on CIFAR10, for example. Although these filters are less clean than those learned on the original Sparsenet data (Olshausen & Field, 1997), this offers a path to training an end-to-end discriminative classification task in the spirit of (Rolfe & LeCunn, 2013). How do the categorical and part of units of that paper align with the well-tuned and broadly-tuned cells of the visual cortex, if at all?

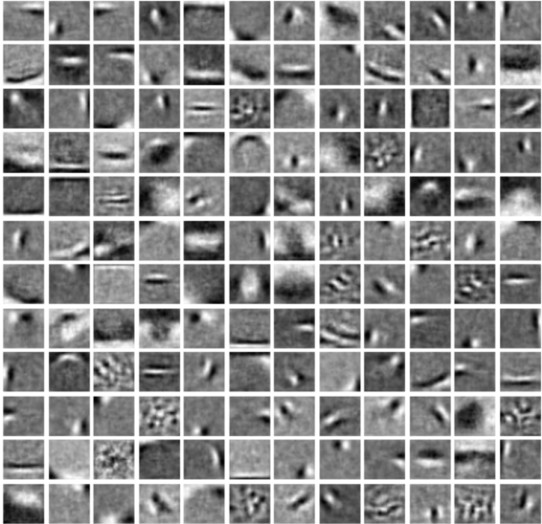

Figure 7: Learned filters for $\lambda = 0.01$ when training the weighted-$\ell_1$ loss on CIFAR10. While these images are $32 \times 32$, the receptive fields still appear to be quite localized in their sensitivity.

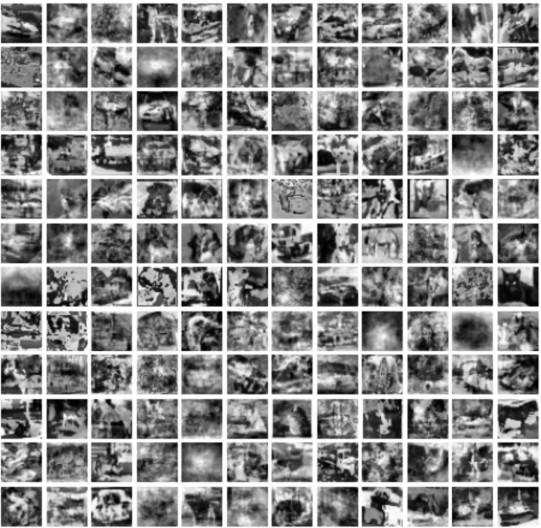

Figure 8: As $\lambda \to \infty$, the filters essentially become mean-prototypes of the various classes in CIFAR, showing that the unrolled weighted-$\ell_1$ can be interpreted as a soft $k$-means objective as well.

