# OpenReview forum: "Local Geometry Constraints in V1 with Deep Recurrent Autoencoders"
_NeurIPS.cc/2022/Workshop/SVRHM — SVRHM Poster_

### Official Review · Reviewer_acso · 2022-10-07
**An interesting, but sparse paper**

**Rating:** 6
**Confidence:** 4

**Review:**

STRENGTHS:

1. This paper discusses an interesting and important question: how can realistic looking receptive fields emerge using sparsity constraints? The authors chose to address this in a different context than what has been recently studied, by making use of spatial regularization. They find that this approach leads to improvements, without having to resort to a different loss function (like LLD).

2. This paper was well written and easy to follow.

3. The figures were clear and well done.

WEAKNESSES:

1. The major weakness, in my view, is that this paper is very short. I understand that this is a workshop paper and it must necessarily be short, but from the SVHRM website, it looks like up to 5 pages is ok. I think the paper could benefit from utilizing this extra space in several ways:
           a. It was not entirely clear to me how this approach differs from LLD. I think I have some idea, and intuition into why it is interesting to look at approaches other than LLD, but it would be good to address this in more detail.
           b. An iterative Laplacian scheme is mentioned as having been tried, but I do not see any results or any discussion of what the results were. Given that this approach differs from your spatial regularization, it would be good to see to what extent it differs in the results. Adding a plot to Fig. 5 and some discussion would make this a stronger paper.

2. This is a personal pet peeve, but I think all figures and appendices should be referenced in the text. Otherwise, it is unclear what I am looking at and what additional information is present in the appendices.

3. The title of this paper does not feel, to me, to be representative of what the paper actually does. As you say in your paper, you do not try to create a one-to-one mapping with visual cortex, so why is V1 mentioned in the title?

OVERALL:

I think the core result is interesting. I think it is, therefore, worth presenting at the workshop, therefore I am recommending it be accepted. However, I think in order for it to be a good paper, the weaknesses highlighted above need to be addressed.

---

### Official Review · Reviewer_srGy · 2022-10-14
**Promising research path but still in preliminary state**

**Rating:** 7
**Confidence:** 4

**Review:**

This paper contains several original contributions :
- Uses a deep autoencoder as the model of the V1.
- Uses L1 regularization to impose locality constraints.
With this framework, they manage to reproduce experimental results at the same level that SOTA while maintaining
the fundamental concepts of sparse coding.

The writing of the paper is unclear.

My main concern with this work is that the experiments appear to be quite preliminary, and they are not adequately described in the paper:
- The description of the data selection is not described in depth in the paper or the appendix, and there is no mention of how this data compares with the one used by the method used as benchmark LLD
- It is complicated to compare the results in figure number 5 with the ones in figure 4. I would recommend to over plot them.
-There is no description of error treatment in the results.
- The only experiment with real images is in the appendix, and it is done with a toy dataset (CIFAR-10) and is only benchmarked against the Sparse algorithm.

In summary, it is an interesting paper containing several important original contributions. More work is needed to clarify the text and to improve the experimental approach. It k is a good paper that should be accepted, and I look forward to seeing future work on this topic.

---

### Official Review · Reviewer_GsGd · 2022-10-14
**Interesting model explaining the diversity of gabor profiles found in V1 - unclear exposition**

**Rating:** 6
**Confidence:** 2

**Review:**

Summary: Multiple existing models such as sparse coding and ICA can account for the emergence of gabor-like filters in the early visual system, but these models usually do not reflect the diversity of gabor profiles found in the visual system. Here the authors propose an architecture and loss function able to reflect this diversity better.

Clarity concerns:
- Although the model proposed seems interesting, it is not clearly exposed. This work builds on an existing model (LLD) which is not described in sufficient detail, and the only reference to this existing model is a COSYNE abstract. In particular, I did not understand what the "local ensembles" and "ensemble members" correspond to wrt the image decomposition into patches.
- It is unclear to me that the model proposed (fig 5) better reflects Gabor spatial phases of rhesus macaque (fig3a) compared to the existing model LLD (fig3b), so it is unclear to me which improvement the authors are referring to. Also, it is unclear to me what is meant by the claim that the model "vastly improves symmetry".

---

### Official Review · Reviewer_9DSa · 2022-10-14
**This paper injects an additional regularization term to sparse coding to induce basis functions that are more similar to biological counterparts.**

**Rating:** 4
**Confidence:** 2

**Review:**

Quality: average
Clarity: below average
Originality: average
Significance: below average

On top of the $\ell_1$ regularization, the authors proposed to add an additional regularization term (4) that matches the basis function to the incoming stimuli, weighted by the firing rate of the basis function. This regularization term is applicable both with the cost function being the input reconstruction in the original compressive sensing paper and with the newer "patches of input" reconstruction. This paper can be of interest if one wishes to model "a layer of" a biological neural system with sparse coding.

---

### Official Review · Reviewer_FeRe · 2022-10-17
**Interesting approach to creating sparse coding filters that are more like those in V1**

**Rating:** 6
**Confidence:** 3

**Review:**

Summary:
In this work the authors recover Gabor-like filters which are more closely aligned with those found in V1 than the filters obtained through the standard sparse coding paradigm (reconstruction loss + l1 penalty on natural image patches). The authors do this by replacing the standard l1 penalty with a “weighted-l1” penalty (WL) where weights are given by the squared Euclidean distance between the learned filters and input images. The authors use algorithm unrolling with an RNN architecture for optimization. The authors also allude to a connection with manifold learning by offering an interpretation of their WL penalty in terms of bipartite graph Laplacians.

Pros:
- Modifying the sparsity penalty to push the learned filters closer towards those found in V1 is an elegant approach.
- Connecting the WL penalty to a bipartite graph Laplacian is an interesting perspective for this sparse coding variant. It would be nice to see the graph embedding/clustering alluded to on lines 99-100.
- The visualizations are insightful and compelling.

Cons:
- Besides the similarity to the spectral embedding loss function, it is not very clear what motivates the design of this WL penalty, especially when considering that the goal is to improve alignment with the filters found in V1.
- Only the fitted phases $\phi$ are explicitly numerically compared to V1 and other methods (Figure 3). It would be helpful to know how other parameters fare.
- The need for the recurrent unrolled architecture could use more explanation. I suspect this was to improve the training speed/convergence. It would be helpful to know how well this approach performs when optimizing via non-DL approaches, or why classical approaches are not sufficient.

Minor Comments:
- The penalty terms are denoted by $S_\lambda^{name}$ (lines 60, 61, 118,  equation 4, 5) but there is no need for the lambda in the subscript as it never occurs in the definition.
- As written, $S_\lambda^{WL}$ is not quite a weighted-l1 norm as there is no absolute value applied to the components of the input vector $x$. I suspect this doesn’t make a difference in the optimization as it seems $x$ is projected to the probability simplex. However this is unclear at first glance, and if that is the case, there is no harm in including the absolute values.
- In equation 4, a matrix $X$ is passed as an input but on the RHS the sole $x_j$ does not depend on $i$. I suspect this was intended to be an $x_{ji}$ (i.e. the j-th row of the i-th column). Indeed, $x_{ij}$ appears later (on line 99).
- On line 94-95 the reader is told that the deep recurrent autoencoder and the projection mapping $P_S$ will be "described below", but then this is not described in the main body. A reference to figure 4 or appendix A would clarify where to look.